# Physical Rehabilitation of Motor Functional Neurological Disorders: A Narrative Review

**DOI:** 10.3390/ijerph20105793

**Published:** 2023-05-11

**Authors:** Ayelet N. Kelmanson, Leonid Kalichman, Iuly Treger

**Affiliations:** 1Department of Physical Therapy, Recanati School for Community Health Professions, Ben-Gurion University of the Negev, Beer Sheva 84105, Israel; 2Department of Rehabilitation, Soroka Medical Center, Beer Sheva 84105, Israel; 3Department of Medicine, Faculty for Health Sciences, Ben-Gurion University of the Negev, Beer Sheva 84105, Israel

**Keywords:** functional neurological disorders, functional motor disorders, conversion disorder, physical rehabilitation, medical care

## Abstract

Functional Neurological Disorders (FNDs) are one of the most common and disabling neurological disorders, affecting approximately 10–30% of patients in neurology clinics. FNDs manifest as a range of motor, sensory, and cognitive symptoms that are not explained by organic disease. This narrative review aims to assess the current state of knowledge in physical-based rehabilitation for motor/movement FNDs in the adult population, with the goal of improving research and medical care for this patient population. To ensure optimal outcomes for patients, it is critical to consider several domains pertaining to FNDs, including which field of discipline they should belong to, how to investigate and test, methods for rating outcome measures, and optimal courses of treatment. In the past, FNDs were primarily treated with psychiatric and psychological interventions. However, recent literature supports the inclusion of physical rehabilitation in the treatment of FNDs. Specifically, physical-based approaches tailored to FNDs have shown promising results. This review utilized a comprehensive search of multiple databases and inclusion criteria to identify relevant studies.

## 1. Background

One of the oldest and most recognized functional disorders is Functional Neurological (Symptom) Disorder (FNSD or FND). Despite the advancement in knowledge and treatment of FNDs within the last 10 to 15 years, clear guidelines for physical-based rehabilitation for this common and debilitating disorder are lacking. This narrative review aims to assess the current knowledge in physical-based rehabilitation for specifically motor/movement FNDs, recognized as FMDs, within the adult population.

## 2. Methods

A narrative review of the current literature on physical rehabilitation in FMDs was conducted using the PubMed database. The search terms “Functional Movement Disorder”, “Functional Motor Disorder”, “Motor Functional Neurological Disease”, “Movement Functional Neurological Disease”, “Rehabilitation”, and “Physical Rehabilitation” were used to identify relevant literature published from 2014 to September 2022. Additional publications were identified from the references cited in the initial papers. Only literature in English was included, and clinical trials were prioritized. The focus was on the most current literature related to the DSM-V’s revised definition of FND, specifically FMDs.

After a thorough review, 48 sources were included, comprising 6 review articles, 4 recommendation articles, 11 clinical trials and case studies, 1 overview-case study, and information gathered from the FND-COM (Functional Neurological Disorder-Core Outcome Measures) and the Functional Neurological Disorder Society. The sources included articles reporting current overviews of FMDs physical rehabilitation, clinical trials, case series, and recommendations.

## 3. Results

### 3.1. Definition and History

FND has a rich historical past and was first documented over 4000 years ago in Egyptian writings, recognized then as *hysteria* [1]. In FND, the worlds of neurology, psychiatry, and psychology collide [2]. Throughout time, the psychiatric and medical fields have grappled to understand the etiology and pathophysiology behind FNDs [3,4]. As the understanding of FNDs has morphed through time, so have the definitions and terminology. Texts dating back to 1900 BC of ancient Egypt, Plato’s writings, and the later *Corpus Hippocraticum* of Greece, are the first sources to describe FNDs in detail, defining the uterus as the organ of origin [5,6]. The Greek word *hysteria* is linked to the uterus in various ways; therefore, the clinical term coined was *hysteria* [5]. Additional Greek sources recount that the disease may seldom appear in males due to sexual organ dysfunction [6]. Beliefs morphed with time, and for an extended period, *hysteria* became attributed to witchcraft and supernatural sources [6]. From the 16th and 17th centuries, the approach shifted once again, and the focus of the disorder moved toward the brain. Sources account for the brain as a potential trigger, and it simultaneously dawned that males may equally suffer from the disease [5]. The disorder once recognized as *hysteria,* became classified as a neurosis, a disorder of the nervous system [6]. In the 19th century, physicians continued to investigate the disorder. Two leading physicians, James Paget and Jean-Martin Charcot, stated in their writings that the disease is of organic origin [7,8]. Charcot initially attributed the symptoms to lesions of the nervous system, and this view transformed throughout his career. From the 1880s, Charcot began to weave psychological terms into his writings, therefore making some of works seem contradictory [7]. The approach that FNDs were of psychological mechanisms was further developed in the late 19th and 20th centuries, as appears in Sigmund Freud and Josef Breuer’s works [6]. Terms such as *conversion*, *(psycho)somatization*, and *dissociation* became directly associated with the condition and later labeled terms. In the first edition of the Diagnostic and Statistical Manual for Mental Disorders by the American Psychiatric Association, the disorder appears in psychophysiologic autonomic and visceral disorders. The conversion reactions of *psychosomatic disorders* were a result of the autonomic nervous system, which innervates organs, and according to this, it did not involve perception or voluntary control and required a psychological stressor [6].

Toward the end of the 20th century, the focus that had been on etiology and the theory behind the disease turned toward symptomatic descriptions. The term *hysteria* was replaced with *somatoform disorder*, *body dysmorphic disorder*, *dissociative disorder*, and *conversion disorder* [6]. The most recent revision of the term and definition, “Functional Neurological Symptom Disorder” (FND), initially appeared in the Diagnostic and Statistical Manual of Mental Disorders, Fifth Edition (DSM-V) [9].

FND is a common and disabling condition associated with motor and sensory symptoms which are incompatible with a recognized neurological or medical condition [9,10] but without neurological functioning signs [9]. The most recent findings support that changes in motor circuitry exist on a neurobiological level [6]. The changes are evident by increased limbic–motor network connectivity and alterations in prefrontal systems that are believed to be related to emotional regulation and cognitive control [11]. Many symptoms are associated with FND, and today, the disorder includes four entities: functional seizures, functional movement disorders, persistent perceptual postural dizziness, and functional cognitive disorder [12]. Symptoms vary greatly depending on the FND subtype. They may include functional limb weakness, dystonia/abnormal movements, tremors, dissociative seizures, urinary retention (with a negative cauda equina syndrome scan), functional cognitive symptoms, speech difficulties, sensory symptoms, dizziness (often recognized as persistent postural perceptual dizziness, PPPD), and visual or hearing signs [3,12]. FND patients commonly suffer from comorbid symptoms such as depression, anxiety, fatigue, bladder and bowel dysfunction, and fibromyalgia [3,12,13]. It is unclear, however, whether to relate these signs as intrinsic characteristics of FNDs, common comorbidities, or sequelae [12]. In the most recent 11th edition of the World Health Organization’s *International Classification of Diseases* (ICD-11) functional disorders, these disorders appear in both neurological and psychiatric categories [14].

Substance abuse (including medications) must be ruled out, clinical and laboratory findings cannot be explained by recognizable neurological or medical conditions, and FNDs should not be accompanied by distress and/or excessive thoughts, feelings, or behaviors and should be distinct from intentionally produced or contrived symptoms [9]. This has better defined FND as a condition stemming from pathophysiological and neurobiological sources, as opposed to psychological anomalies [15]. The symptoms of FND are generally incapacitating and become a significant source of distress to the patient [4]. Patients presenting with FND are frequently labeled as “difficult patients”; upon evaluation, they are blamed for their disorders and are often dismissed by specialists [10].

### 3.2. Epidemiology and Prevalence

Since the revision of FNDs’ terms and definitions in the DSM-V, the condition’s description has become significantly clearer, thereby substantially improving the accuracy of rates and risks in the most current literature [15]. Recent findings support that a 6% occurrence rate of FNDs exists within all outpatient and community neurological contacts, or 4 to 12 cases for every 100,000 patients in primary care per annum [4]. This makes FND the second most common reason for referral to neurology [13] and one of the most common causes of neurological disability [15]. The rate of diagnosis appears to be consistent with imagery findings in computed tomography and magnetic resonance imaging, which is reliable with a calculated revision rate of merely 5% [4]. Furthermore, in studies with a five-year follow-up, the rate of misdiagnosis was only 4% [15]. It is believed that twice as many neurological patients are misdiagnosed in the opposite direction [3].

There is a significantly higher occurrence of FNDs in women (3:1 ratio) [3,4,15]. Diagnoses peak within the 35 to 50-year age gap, yet they do exist throughout the lifespan; pediatric (rarely seen before the age of 10), adolescent, and geriatric patients in their 80 s are seen within the medical system [3,4,15]. The estimated healthcare costs for patients with FND are USD 900 million annually in the United States of America [3,10], and a significant percentage of outpatient neurological care is attributed to FND patients [3].

### 3.3. Mechanism and Etiology

FND was once considered a somatic or conversive disorder, believed to result from precipitating stressors of trauma or psychiatric disorders [10]. Recent studies fail to demonstrate this link, leading to the DSM-V no longer requiring a presence of existing stressors for the diagnosis of FND, despite the lack of a known organic source [9,10,15]. The most current theories of the pathophysiological basis of FNDs are based on recent functional neuroimaging studies [10]. Recent imagery studies demonstrate that the dysfunction in FNDs manifests at a brain network level and involves areas responsible for attention, emotion, and motor planning. The current model explains pathophysiology as the overactivity of the limbic system and the prefrontal cortex. It seems that deficiencies in the coding framework coupled with overactivity in other regions create an internal symptom model responsible for voluntary predictions and physical sensations. In functional imagery studies, activity patterns involved in FNDs have been located across the entire brain, making it difficult to determine specific paradigms [16]. The results of functional imagery studies vary based on the specific FND symptoms and the method of investigation [17]. The basal ganglia and the cerebellum were found to be involved in functional dystonia [18]; functional connections between the Supplementary Motor Area and altered activity in the amygdala were apparent in patients with motor FNDs with the use of emotional stimuli [19,20,21,22]. In the study published by Voon et al., findings also reflect hypoactivity in the right and lower temporoparietal junction, coupled with lower functional connectivity in sensorimotor (sensorimotor cortices and cerebellar vermis) and limbic regions (ventral anterior cingulate and right ventral striatum) [21]. These findings suggest that sensory prediction signals and proprioceptive feedback in FND patients are significantly altered, and further the belief that conversion movements are not voluntary and self-generated. Additional imagery studies do similarly indicate that attention and sensory perception of voluntary movements may be directly involved in FND symptoms [17,23,24,25]. Another common feature found in fMRI studies conducted on motor FND patients is altered activity in the insula, one of the complex multi-functional regions deeply involved in self-awareness and emotional regulation [19,22,26,27].

Following these findings, one may conclude that it is difficult to explicate a specific model of the neural paradigm of functional motor disorders due to the broad and various brain regions involved. One may summarize that FND patient experiences misinterpretation of sensory input, which thus interferes with motor planning [10,12]. Depending on the symptom manifestations, the functional variants have presented in the following locations: the prefrontal cortex (when voluntary movement is compromised) and specifically the ventromedial prefrontal cortex, the supplementary motor area (in tremor, dystonia, or gait abnormalities), temporoparietal junction (with functional tremors), and the primary amygdala and cingulate gyrus (associated with self-awareness and emotion recognition) [2,15].

### 3.4. Treatment

Many FND patients have negative experiences during treatment within the healthcare system. The negative experiences are generally the result of a combination of poor knowledge on the part of healthcare providers and a tendency to disbelieve, blame, or shame patients suffering from FND. The approach adopted by physicians often results in statements that the symptoms do not exist or accusations that the patient has voluntarily created the symptoms, and the course taken thereafter is psychiatric care [3]. In the past, the treatment approach to FND was mostly inpatient psychiatric care [1]. However, from the 1980s onward, it is evident from the minimal literature that exists on conversive disorders that the therapeutic approach began to evolve [28,29,30,31,32]. The approach became multidisciplinary, including inpatient physical rehabilitation.

The balance of treatments in motor FNDs is complex for the following reasons: physical and psychological symptoms are comingled, symptoms vary significantly between patients, and unique aspects point towards relating to subjective measures (as opposed to most other disorders that calculate objective clinical measures) [13].

Limitations also derive from the fact that comparatively little research has been conducted in the past. However, this has significantly changed within the last 10 to 15 years as increasing awareness of FNDs has compelled further research [10,13,15]. Currently, the adopted method for FND medical care is a multidisciplinary-integrative approach which is implemented by a team of varying specialists [3,10]. The team framework ensures consistent messages with coordinated treatments, eliminating unnecessary testing and referrals to multiple specialists [10]. In the treatment plan, a great emphasis is put on communication with the patient. A clear explanation of the diagnosis is paramount, together with an assessment of the patient’s response in a manner that reflects their understanding of the diagnosis [3,12]. The majority of patients suffering from FND in care are specifically motor FND (Functional Motor Disorders, or FMDs), and FMD patients tend to present with severe symptoms [13]. The treatment plan generally includes the management of physical symptom comorbidities, physiotherapy, psychiatry, psychological therapy, occupational therapy, speech, and language therapy, medication, hypnosis, and other treatments (such as inpatient care and therapeutic sedation), all based on symptom manifestation and the patient’s specific needs [3,12].

## 4. Discussion

One may draw the conclusion that the literature regarding physical rehabilitation for FMD has significantly increased within the last decade. Research on FMDs has increased, and the establishment of the Functional Neurological Disorder Society (FNDS) in 2019 has signified critical markings with regard to the condition [13]. From the early 21st century, the recommended treatment strategy for FMD supports a multidisciplinary approach of behavior modification (commonly cognitive behavioral therapy), psychotherapeutics, and physical rehabilitation [1,10]. Motor rehabilitation is administered by physiotherapists, occupational therapists, and speech therapists who are familiar with and trained explicitly in FMD rehabilitation [15]. The goal is to restore normal movement patterns, thereby regaining patient independence [33]. Motor retraining begins with elementary movements and progresses toward more complex movements as the patient improves [33]. A therapist may incorporate visual or electromyography biofeedback, transcranial magnetic stimulation, or treadmill training [33].

Consensus recommendations have been published; however, due to the heterogeneity of symptoms in FMD patients, it is challenging to recommend uniform interventions [33,34]. Therefore, there are no specific guidelines for physical rehabilitation and treatment of FMDs to date. Physical rehabilitation generally includes (psycho)education, progressive movement/motor retraining (MoRe), behavioral strategies (recognizing and praising achievements), and encouragement of long-term maintenance [10,35]. A unique approach has been adopted in the physical rehabilitation of FMD. As opposed to the treatment of most neurological conditions, in which the therapist draws attention to the affected area or limb, in FMDs the approach is to encourage natural or automated movements in a functional setting while minimizing the attention paid to the symptom or affected limb [3,10,34,35]. Although this therapeutic approach may seem paradoxical, it has proven to be significantly more effective than standard physical therapy [3,36]. In a study published by Nielsen et al., participants in the unique motor FND physical rehabilitation program showed a 72% improvement rate in symptoms following six months of participation in a one-week unique-FND physical rehabilitation program, compared with participants in standard physiotherapy, who exhibited a 28% improvement rate at six months after rehabilitation [36]. There are no specific recommendations regarding the frequency and length of recommended rehabilitation, yet studies with intense physical therapy protocols require a minimum of five treatments per week, even if for only one whole week. These studies have reported substantial results, which are also maintained long-term [33,37,38,39,40].

Another aspect that was lacking in the field of motor FNDs, due to the complexity and heterogeneity of the disorders, was a clear set of outcome measures [13]. Over the years, optimal outcome measures have become recognized as key components of evidence-based medicine for treatment and clinical trials. Up until the establishment of the FND-COM, which is a group of 45 professional members who initially met in September 2017 (Edinburgh, UK) and subsequently in September 2019 (Atlanta, GA, USA), there were no clear outcome measures for FNDs (FND-COM, COMET initiative, 2017). The result of the work of the FND-COM group was a set of clear outcome measures, which includes, to-date, five FND-specific measures (three are clinician-rated, and two are patient-rated), with no single measure identified as a cross-range symptom for adults with FND (FND-COM, COMET Initiative, http://www.comet-initiative.org (accessed on 8 May 2023)). Additional commonly measured domains are life impact, health economics/utility costs, and physical and psychological symptoms (FND-COM, COMET Initiative, http://www.comet-initiative.org (accessed on 8 May 2023)).

Physical rehabilitation for motor FNDs can be in an outpatient, inpatient, or telehealth setting, depending on the nature of symptom onset [35,41,42]. When patients present with acute FMD, they often initially appear in emergency care, are subsequently hospitalized, then 20% continue to inpatient rehabilitation [35,43]. Inpatient care offers the benefits of frequent and coordinated treatment by a multidisciplinary team in a protective and sheltered setting [35]. Outpatient interventions are suitable for patients with less severe symptoms; the patient is in his/her familiar environment while they continue with normal activities of daily living; thereby, the outpatient applies therapeutic strategies functionally. The benefits of outpatient therapy can also be seen with short-term care [15,36,40]. Telehealth has recently become fairly common, and studies have reported positive outcomes with statistically significant measures [41,42]. Telehealth rehabilitation begins with several (generally three) initial assessment and therapy sessions, which are face-to-face, while the remaining sessions continue in a virtual setting. The positive aspects of telehealth are that it is readily available, accessible, and primarily followed through in the patient’s natural, functional home environment. Telehealth thereby enables longer-term care for patients as well.

The main principles for the physical rehabilitation of FMDs are concluded in Table 1.

## 5. Conclusions

FNDs are complex neurological disorders that require a multidisciplinary approach to management. While the traditional approach of psychological and psychiatric interventions has been the norm, recent literature suggests that physical rehabilitation should be a crucial part of the management of FNDs. The literature reviewed in this narrative review provides evidence of the effectiveness of physical rehabilitation in improving motor function and quality of life in individuals with FNDs. However, further research is needed to establish optimal rehabilitation protocols, including the intensity, frequency, and duration of physical therapy interventions. In addition, the development of standardized outcome measures and the implementation of interdisciplinary care will help improve the accuracy of FND diagnosis and management. Despite these challenges, the integration of physical rehabilitation in FND management provides hope for individuals with FMDs, particularly those who are limited in their daily activities due to the disorder.

In addition, the study identifies three settings for physical rehabilitation: inpatient rehabilitation, outpatient care, and telehealth, which can be tailored to the specific needs of individual patients. As research and literature continue to emerge on the topic, it is important to explore and identify more specific and effective interventions in physical rehabilitation. This will lead to better outcomes and higher quality of life for patients with FNDs. Ultimately, the most effective treatment approach for FNDs is one that includes physical rehabilitation with a unique and tailored approach.

## Figures and Tables

**Table 1 ijerph-20-05793-t001:** General treatment and therapy principles for physical rehabilitation for FMDs.

**Physical Rehabilitation Setting**
Inpatient
Outpatient
Telehealth
**Physical Rehabilitation Approach**
* Physical rehabilitation exists in a biopsychosocial framework with a multidisciplinary team.
Followed through in a functional setting.
Encouragement of natural or automated movements.
Minimization of the attention paid to the symptom or the affected limb.
**Therapy Methods or Strategies**
Establishment of clear treatment goals: relearning motor control.
(Psycho)Education and behavioral strategies.
Emphasis on quantity of movement versus quality of movement.
Progressive movement/motor retraining (MoRe).
Treatment adjuncts (i.e., electrical stimulation, biofeedback, and transcranial magnetic stimulation) may be beneficial.
Long-term maintenance.
**Frequency & Duration of Rehabilitation**
A minimum of five treatments per week for substantial results in inpatient care.
One-week rehabilitation programs for outpatient care have shown substantial improvement with long-term results.
A minimum of 24 telehealth sessions (initial 3 face-to-face, 21 tele-sessions) are shown as beneficial.
**Outcome Measurement**
3 clinician-rated scales [44,45,46]
2 self-report measures [47,48]
Currently, no measures specifically for functional limb weakness or sensory symptoms exist.
There are currently no single outcome measure suitable for use across all adult FND symptom types.

## Data Availability

Not relevant (study is a narrative review).

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
