# Peer review of "Physical Rehabilitation of Motor Functional Neurological Disorders: A Narrative Review"

_ijerph, 2023, doi:10.3390/ijerph20105793_

Round 1
Reviewer 1 Report
Review
The aim of this narrative review is to assess the most current knowledge on physical-based rehabilitation of motor/movement FNDs in the adult population. As the authors assess, therefore, it is difficult to determine several domains pertaining to FNDs . In FND, the worlds of neurology, psychiatry, and psychology collide. As the understanding of FND has morphed through time, so have the definitions and terminology. FND is a common and disabling condition associated with motor and sensory symptoms which are incompatible with a recognized neurological or medical condition. Recent findings confirm that a 6% incidence of FND exists in all ambulatory and community neurological contacts. The rate of diagnosis appears to be consistent with imagery findings in computed tomography and magnetic resonance imaging, which is reliable with a calculated revision rate of merely 5%.
In the studies, you write that there are still no specific guidelines no specific guidelines for physical rehabilitation and treatment of FMDs to date. As you point out, Nielsen et al., participants in the unique motor FND physical rehabilitation program showed a 72% improvement rate in symptoms following six months of participation in a one-week unique-FND physical rehabilitation program, compared with participants in standard physiotherapy, who exhibited a 28% improvement rate at six months after rehabilitation.
Conclusion
It is also suitable for outpatients. An outpatient can apply functional therapeutic strategies using telehealth, which is easily available, accessible and primarily monitored in the patient's natural, functional home environment.
Questions:
1. Recent studies demonstrate that the dysfunction in FNDs manifests at a brain network level and involves areas responsible for attention, emotion, and motor planning. Have you tried looking for sources of knowledge in this area, for example, also in energy medicine?
2. When looking for information on this topic, don't you think about other alternatives that are more tried and tested? For example, in Eastern medicine?
Author Response
Questions:
- Recent studies demonstrate that the dysfunction in FNDs manifests at a brain network level and involves areas responsible for attention, emotion, and motor planning. Have you tried looking for sources of knowledge in this area, for example, also in energy medicine?
Reply: In this narrative review, we specifically focused on the role of physical rehabilitation of Motor FNDs. The authors are physical therapists and rehabilitation physicians, and we were interested in looking for recent knowledge that can lead us to inpatient rehabilitation of patients with FNDs. The subject that reviewers suggested is interesting but requires a separate review.
- When looking for information on this topic, don't you think about other alternatives that are more tried and tested? For example, in Eastern medicine?
Reply: Please see the answer to the previous comment. In a clinical setting, the methods of Eastern medicine are used as a part of complex integrative treatment. However, this subject was not part of this review's aims.
Reviewer 2 Report
The paper has potential, but every part seems unfinished and unclear. In the first part /Definition/, I suggest to further develop the historical transformation of the term /explain hysteria in details/ and examine the modern concept in the group of “Somatic Symptom and Related Disorders”, presenting the symptomatology and criteria that are necessary for the diagnosis Functional neurological symptom disorder (conversion disorder). The link between both is also unclear /the authors must underline that they are equal and clarify the term conversion/. A good idea would be to mention the analytical point of view, since it is a basic model for understanding the “conversion” process in general. The inclusion of the word "motor" must also be explained, although in the Definition part I can not understand if it is the same disorder.
I would suggest to explain the group and to mention the other disorders in brief Somatic symptom disorder and Illness anxiety disorder. However, Functional neurological symptom disorder (conversion disorder) is a medical disorder that is placed under the classification of mental disorders. Talking about classification systems, the authors must also include the other popular classification – ICD.
In the third part Mechanism and etiology I think that the authors must include more details and explain the possible connection between the brain parts mentioned. I want to see the authors’ examples of possible mechanisms, for ex. what happens first, second and third if the amygdala and cingulate gyrus /which part?/ are involved? And is that evidence of a biological base of the disorder? If the emotional control is involved what can happen with the prefrontal brain areas ….. /the connection between ACC, limbic structure and OFC/. Attention control is a good example in the Therapy part.
Of course, this would be a suggestion of the mechanism only, but if the authors stand for the idea of a biological mechanism together with some psychological reactions, then we would expect to find at least an idea of a model.
The methods must be explained in details and clearly – I can not understand the selection criteria.
The authors must remove all the discussion part in the Results part and present their findings in tables or some other visual way.
The discussion part must be rewritten. It must contain a discussion – the authors’ explanation of the results and comparison /references/ with other findings /if possible/.
The conclusion is very short.
Finally, the paper is very short, and 26 references are not enough for a review. I suggest to include at least 50-60 references.
Author Response
The paper has potential, but every part seems unfinished and unclear.
Reply: We thank the reviewer for the detailed and important comments. We tried our best to answer all the comments in a revised draft.
In the first part /Definition/, I suggest to further develop the historical transformation of the term /explain hysteria in details/ and examine the modern concept in the group of “Somatic Symptom and Related Disorders”, presenting the symptomatology and criteria that are necessary for the diagnosis Functional neurological symptom disorder (conversion disorder). The link between both is also unclear /the authors must underline that they are equal and clarify the term conversion/. A good idea would be to mention the analytical point of view, since it is a basic model for understanding the “conversion” process in general.
Reply: Following the reviewer's suggestion, we added an extensive part on the historical transformation of the term to the Definition section of the paper (pp. 3-4).
The inclusion of the word "motor" must also be explained, although in the Definition part I can not understand if it is the same disorder.
Reply: We added the explanation to the Definition section of the paper (p. 4).
I would suggest to explain the group and to mention the other disorders in brief Somatic symptom disorder and Illness anxiety disorder. However, Functional neurological symptom disorder (conversion disorder) is a medical disorder that is placed under the classification of mental disorders. Talking about classification systems, the authors must also include the other popular classification – ICD.
Reply: We added this information on p. 5.
In the third part Mechanism and etiology I think that the authors must include more details and explain the possible connection between the brain parts mentioned. I want to see the authors’ examples of possible mechanisms, for ex. what happens first, second and third if the amygdala and cingulate gyrus /which part?/ are involved? And is that evidence of a biological base of the disorder? If the emotional control is involved what can happen with the prefrontal brain areas ….. /the connection between ACC, limbic structure and OFC/. Attention control is a good example in the Therapy part.
Reply: Following the reviewer’s suggestion, we extensively expanded this part of the paper (pp. 6-8).
Of course, this would be a suggestion of the mechanism only, but if the authors stand for the idea of a biological mechanism together with some psychological reactions, then we would expect to find at least an idea of a model.
Reply: The main purpose of this review is not to suggest the mechanisms of FNDs, but review and organize the rehabilitation options, with an accent on physical rehabilitation. Therefore, presenting the model of FNDs is not in the scope of this review.
The methods must be explained in details and clearly –I can not understand the selection criteria.
Reply: In a narrative review, the method of the literature search is not necessary. We included this part to show the keywords used, so if someone plans a further study (for example, systematic review, it will help to find relevant literature). To make the methods clearer, we rewrite it.
The authors must remove all the discussion part in the Results part and present their findings in tables or some other visual way.
Reply: We added the Table that concludes our findings at the end of the discussion.
The discussion part must be rewritten. It must contain a discussion – the authors’ explanation of the results and comparison /references/ with other findings /if possible/.
Reply: The discussion part of our review combines and organizes the results presented in precious parts of the review. We also added the Table of main findings.
The conclusion is very short.
Reply: We extended the conclusion part (pp.13,14).
Finally, the paper is very short, and 26 references are not enough for a review. I suggest to include at least 50-60 references.
Reply: This review is defined as a narrative review, and there is no specific requirement for the number of references in this type of review. We used all the relative literature that we found on the specific topic. Answering the reviewer's comments, we added 22 references to the paper, resulting in 48 references. We believe that the revised paper is comprehensive enough and hope it can be accepted in a present form.
Reviewer 3 Report
It is necessary to make a search flowchart of articles in the area and place it in the results. It would also be important to place a table with the main findings of each article to guide the reader. The subject is super relevant, but adjustments are needed in the methods and adding the results.
Author Response
It is necessary to make a search flowchart of articles in the area and place it in the results.
Reply: As mentioned in the paper's title and Methods section, our paper presents the narrative review. In this type of paper, the search flowchart is not required.
It would also be important to place a table with the main findings of each article to guide the reader.
Reply: We added the Table that concludes our findings at the end of the discussion.
The subject is super relevant, but adjustments are needed in the methods and adding the results.
Reply: We wish to thank the reviewer for finding our paper relevant. We addressed all the reviewer’s comments in a revised draft.
Round 2
Reviewer 2 Report
No other suggestions
Reviewer 3 Report
The work is suitable for publication.